# Multifunctional Layered Double Hydroxides for Drug Delivery and Imaging

**DOI:** 10.3390/nano13061102

**Published:** 2023-03-19

**Authors:** Seungjin Yu, Goeun Choi, Jin-Ho Choy

**Affiliations:** 1Department of Nanobiomedical Science and BK21 PLUS NBM Global Research Center for Regenerative Medicine, Dankook University, Cheonan 31116, Republic of Korea; 2Intelligent Nanohybrid Materials Laboratory (INML), Institute of Tissue Regeneration Engineering (ITREN), Dankook University, Cheonan 31116, Republic of Korea; 3College of Science and Technology, Dankook University, Cheonan 31116, Republic of Korea; 4Division of Natural Sciences, The National Academy of Sciences, Seoul 06579, Republic of Korea; 5Department of Pre-Medical Course, College of Medicine, Dankook University, Cheonan 31116, Republic of Korea; 6International Research Frontier Initiative (IRFI), Institute of Innovative Research, Tokyo Institute of Technology, Yokohama 226-8503, Japan

**Keywords:** layered double hydroxide, inorganic nano vehicle, drug delivery, bio-imaging, therapy, diagnosis, theranostics

## Abstract

Two-dimensional nanomaterials, particularly layered double hydroxides (LDHs), have been widely applied in the biomedical field owing to their biocompatibility, biodegradability, controllable drug release/loading ability, and enhanced cellular permeability. Since the first study analyzing intercalative LDHs in 1999, numerous studies have investigated their biomedical applications, including drug delivery and imaging; recent research has focused on the design and development of multifunctional LDHs. This review summarizes the synthetic strategies and in-vivo and in-vitro therapeutic actions and targeting properties of single-function LDH-based nanohybrids and recently reported (from 2019 to 2023) multifunctional systems developed for drug delivery and/or bio-imaging.

## 1. Introduction

Numerous two-dimensional (2D) nanomaterials, including layered double hydroxides (LDHs) [1,2,3,4,5,6,7,8,9,10], silicate clays (montmorillonite, mica) [11,12,13,14], graphitic carbon nitride (g-C_3_N_4_) [15,16], and graphene [16,17], exhibit a size lower than 100 nm, with highly anisotropic and planar architecture [18,19,20]. In the last several decades, their utility in various fields, such as drug delivery [1,2,5,11,12,13], imaging [6,7,8], catalysis [4,15], and sensing [9,10], has been investigated. Among these 2D nanomaterials, LDH delivery-carriers, with biocompatibility, biodegradability, controllable drug release/loading ability, and high cellular permeability, exhibit high potential for biomedical applications such as drug delivery and imaging [21,22]. Intercalative nanohybrids were first reported in 1999 [23]; the paper proposed the utilization of bio-inorganic nanohybrids for gene delivery. Subsequently, LDH-based multifunctional hybrids have been developed and extensively analyzed for application in numerous fields, particularly the biomedical field. 

The lamellar structure of LDH (M_1–x_^2+^ M*_x_*^3+^(OH)_2_]^x+^[A^n−^]_x/n_·mH_2_O) consists of brucite layers that contain divalent (e.g., Mg^2+^, Zn^2+^, and Cu^2+^) and trivalent (e.g., Al^3+^, Ga^3+^, and Fe^3+^) metal-hydroxide layers, with intercalating interlayer anions (A^n−^) to compensate for the metal-layer positive charge. Various anions (such as CO_3_^2−^, NO_3_^−^, and Cl^−^), organic molecules, and drugs have been incorporated in the lattice interlayers of LDHs, depending on their target application. Through 2D lattice engineering, nanohybrids are constructed by stacking 2D host building blocks (LDH layers) with inorganic/organic molecules or biomolecules (forming interlayers) (Figure 1) [22,24,25,26].

As LDH-lattice 2D frameworks are positively charged, negatively charged molecules are predominantly located on their surface or interlayer spaces. Guest species are preponderant in the interlayer; they are incorporated by several synthetic methods. The encapsulated guest species, positioned between the layers, are stabilized by hydrogen bonding and electrostatic interactions. Negatively charged species that are challenging to incorporate via intercalation are adsorbed on the positively charged LDH layers by surface adsorption [30].

Previous studies have demonstrated that LDH nanoparticles can penetrate cell membranes and provide desirable biomedical outcomes. LDH particles smaller than 250 nm can enter cell membranes through receptor-mediated endocytosis, as has been shown by intercellular uptake through clathrin-mediated endocytosis, indicating that LDHs could act as effective vehicles with targeting capabilities through this specific endocytic pathway (Figure 2a) [31,32]. One study revealed that the fluorescein 5′-isothiocyanate (FITC)-LDH nano delivery system, with a size of about 100 nm, could permeate MNNG/HOS bone cancer cells through clathrin-mediated endocytosis, as evidenced by immunofluorescence microscopy and confocal laser scanning microscopy. Further treatment with clathrin antibody and its secondary antibody (Texas Red) confirmed that the intercellular uptake mechanism of LDHs (green fluorescence) was related to clathrin-mediated endocytosis, rather than caveolae-mediated endocytosis (Figure 2b) [31]. Particles larger than 300–350 nm exhibit low cellular uptake owing to their nonselective intercellular uptake route. These findings suggest that particle size and size control are important factors for targeted uptake through site-specific clathrin-mediated endocytosis [31,32].

Among the diverse biomedical applications of LDHs, this review highlights recent progress in LDH-based single-function and multifunctional nanohybrids developed for drug delivery or/and bio-imaging. Additionally, the synthetic strategies and applications of LDH nanohybrids in drug delivery, imaging, and theranostics (a combination of diagnostics and therapeutics) reported in the last five years (2019 to 2023) are discussed, along with their limitations.

## 2. Synthetic Methods for Bio-Applicable Molecule–LDH Nanohybrids

On the basis of their anionic exchange capacity, LDHs have gained interest as a reservoir for negatively charged bioactive compounds. By synthetically manipulating the content and structure of the LDH, it may be possible to produce bio-applicable intercalating moieties between the layers for various biomedical applications. Several chemical methods such as coprecipitation [1,2,33,34,35,36], ion exchange [3,37,38,39], exfoliation-reassembling [39,40,41], and calcination–reconstruction [10,35,42,43,44,45] can be used for lattice engineering to prepare biofunctional LDH nanohybrid materials, as shown in Figure 3 [46,47]. The guest species are accommodated within the interlayers of LDHs through electrostatic interactions, hydrogen-bonding linkages, and van der Waals forces between the guest species and the hydroxyl groups of LDH layers, as well as interlayer anionic and water molecules [21,30]. Therefore, intercalative routes are stabilized, and bio-applicable molecule–LDH nanohybrids can be synthesized and stabilized via various weak interactions.

### 2.1. Coprecipitation Method

The most common technique for preparing drug-LDH nanohybrids is coprecipitation. In this process, the pH of the solution is purposefully controlled to a level where the lattice containing metal (M(II) and M(III)) hydroxides are coprecipitated, with similar pH that deprotonates functional groups of encapsulated molecules, and is achieved by titrating the metal salt solution with an alkaline solution such as NaOH in the presence of intercalating anionic molecules with vigorous stirring [48]. The precipitate generated is easily separated by centrifugation and then washed to eliminate impurities. In addition, hydrothermal conditions can be manipulated to control the nanoscale crystallinity of LDH nanohybrids [49].

Wang et al. used coprecipitation to create Fe_3_O_4_@Mg_3_Al-CO_3_ LDH with a hierarchical core-shell structure as an effective magnetic adsorbent [35]. Similarly, Dilmaghani et al. prepared the Fe_3_O_4_@LDH multi-core@shell system for drug delivery by the same method [34].

Choy et al. used coprecipitation methods to directly prepare drug-LDH nanohybrid systems such as folinic acid-LDH [33], methotrexate (MTX)-LDH [2], and ursodeoxycholic acid (UDCA)-LDH [1]. Additionally, they used LDHs as a boron delivery vehicle for boron neutron capture therapy (BNCT), which requires adequate boron uptake by tumor cells. They hybridized icosahedral boron clusters, such as sodium mercaptoundecahydro-closo-dodecaborate (BSH, Na_2_B_12_H_11_SH), with LDH using the coprecipitation approach by titrating an aqueous solution of Zn/Al mixed salt with NaOH in the presence of boron sources at room temperature. X-ray diffraction (XRD) analysis (Figure 3a) shows that the basal spacing of BSH molecules intercalated in BSH-LDH differs from that of pure LDH [36].

### 2.2. Ion Exchange Method

Another common post-synthetic method is ion exchange, which involves replacing pre-occupied anions with other organic or biological interlayer anions in the LDH interlayer space via concentration gradient to achieve charge balance [30]. First, pristine LDH, which initially contains immobilized exchangeable anions, particularly NO_3_^−^ and Cl^−^, should be synthesized. The LDH precursor is then intercalated with a solution containing an excess of the target anion by changing the pH value. In this reaction, the anion-exchange capacity of LDHs is the determining factor, which depends on layer charge density and its distribution (same as effective layer charge per unit area), as well as the size of the guest ion and particle size [47]. Additionally, it is the preferred method because it prevents the intercalation of unwanted anions while preserving the morphology of the host LDH [50].

Amino acids such as phenylalanine (Phe) and glutamic acid (Glu) were intercalated in each LDH carrier (Phe-LDH, Glu-LDH) to form nanohybrids of amino acids and LDHs using the ion exchange method [3]. Figure 3b depicts the XRD pattern of a single phasic amino acid-LDH containing twice the amount of each amino acid compound compared to coprecipitation methods carried out by other groups.

Meanwhile, the ion exchange approach has a higher crystallinity than other synthetic methods, such as exfoliation-reassembling or calcination-reconstruction for intercalating bioactive compounds such as ferulic acid [39]. Three distinct methods were attempted for ferulic acid intercalation in LDH nanohybrids. Powder XRD studies show that the ion exchange method exhibits superior crystallinity and preserves the crystallite size of pristine LDH.

### 2.3. Exfoliation-Reassembling Method

Recently, the exfoliation-reassembling technique was suggested to encapsulate large-sized or bulky biomolecules [47]. As shown in Figure 3c, a stack of monolayer sheets, LDH, could be separated as a single sheet bearing metal hydroxides by treating the pristine LDH with organic solvents such as formamide to facilitate strong interactions between the highly polarized solvents and the surface hydroxyl groups of LDH [30]. The layers were then reassembled into a layer-by-layer stacking structure or a core-shell structure in a solution capable of dissolving suitable anions. Choy et al. suggested a spherical core@inorganic shell composed of DNA molecules whose negative charge could electrostatically interact with the positive surface of LDH (Figure 3c) [41]. Base-pair duplex DNA was enclosed in exfoliated LDH nanosheets, resulting in a thermodynamically stable state when MH nanosheets were reassembled.

Rather than using coprecipitation and ion exchange reactions, gallic acid (GA) was exclusively produced via exfoliation and reassembling using formamide to form a colloidal solution [40]. The release analysis revealed that manufactured LDHs slowly released, acting as a protective agent in harsh environments such as pH 7.4 at 4 °C and having remarkable antioxidant properties, indicating the suitability of LDH storage from oxidation prior to application.

### 2.4. Calcination-Reconstruction Method

Another intriguing option for creating pharmacological LDH nanohybrids is to use the structural memory effect of LDHs, also known as the calcination-reconstruction method (see Figure 3d). The dehydration and dehydroxylation events, as well as guest ion de-intercalation reactions, cause the disintegration of LDH into mixed metal oxide, amorphous X-ray phase, and a meta-phase when LDH is calcined at high temperatures of approximately 400–500 °C [30,48]. By simply rehydrating and rehydroxylating, formed amorphous metal oxides can be reverted to the original crystalline LDH, giving rise to the topotactic transformation reaction. If such reactions occur spontaneously in the same reaction solution as existing drug moieties, the drug-LDH nanohybrid is synthesized simultaneously, producing an ionic connection. In addition, mesopore production and increased specific surface area are caused by crystal deformation and metal oxide interstratified structure [43]. This technique has been extensively researched to intercalate relatively large or bulky molecules or to avoid intercalating the original anion rather than the desired one [48].

Sipos et al. used several synthetic approaches to intercalate naproxen or diclofenac, nonsteroidal anti-inflammatory medicines, into the interlamellar area of CaFe-LDH for cutaneous administration [45]. Based on the highest organic content and average crystal thickness, the dehydroxylation-rehydration approach was the most suited for drug-LDH composites among coprecipitation, direct anion exchange, dehydroxylation-rehydration, and mechanochemical treatment. In addition, calcination of core-shell structured Fe_3_O_4_@Mg_3_Al-CO_3_ LDH yields an effective magnetic adsorbent to remove the anionic dye C.I. Acid Yellow 219 (AY219) from aqueous solution using the memory effect of LDH [35]. Figure 3d depicts the maximum absorption efficiency due to the immobilization of an anionic dye, AY219, in the interlayer area to replace the carbonate anion once LDH is reconstituted.

## 3. Design of Functional LDH Nanohybrids for Drug Delivery

The primary purpose of drug-delivery systems (DDSs) is to transport and release drugs at the target site for maximum therapeutic efficacy with minimum influence on non-target sites [51]. Drug carriers are used for drugs with in-vivo instability, poor bioavailability, solubility, body absorption, and poor target-specific delivery (adversely affecting non-target sites) [52].

LDHs, with low toxicity, high drug-loading capacity, and the ability to enhance drug bioavailability and solubility, overcoming drug resistance, have been widely studied as DDSs. Moreover, due to the solubility of LDHs in acidic solutions, LDH drug carriers exhibit pH-dependent drug release. The unique structure of LDHs enables facile lattice-composition modification, facilitating the incorporation of different interlayer anions (drugs or drug-assistant molecules) for high therapeutic efficiency. This section highlights LDH-based DDSs with efficient drug release and their in-vitro therapeutic activity, as well as further in-vivo study.

### 3.1. Drug Release Profile (In-Vitro)

The mode of drug release (such as controlled and sustained) is an essential aspect of research on DDSs, as it determines the ability of the system to release the drug concentration that meets the therapeutic requirements. Additionally, it could immensely benefit patients requiring multiple drugs daily. Uncontrolled drug release is initially highly toxic and subsequently ineffective. In contrast, gradual drug release could eliminate the need for several drug administrations in a day.

Several recent studies have analyzed and enhanced the LDH-based delivery of therapeutic molecules [5,53], anti-inflammatory drugs [54,55], anticancer drugs [27,56,57,58,59,60], and peptide hormones [61] by improving the drug-release property of LDHs by modification via intercalation, absorption, and other methods. For example, a publication reported the intercalation of nicotinic acid (NA) in LDH interlayers [5]. NA, with low bioavailability, is one of the oldest and most widely used hypolipidemic drugs; however, the rapid release of NA causes undesirable side effects. To enable the gradual release of NA, it has been encapsulated in LDH and coated with Eudragit^®^ S100 (an FDA-approved enteric coating agent that only dissolves under basic conditions, such as those present in the intestine). To form Eudragit^®^ S100-coated NA-LDH, NA intercalated in LDH (labeled NA-LDH) is synthesized by coprecipitation and spray-dried with Eudragit^®^ S100. As shown in Figure 4a, the concentration of NA increases rapidly within the first few minutes in a simulated gastric solution (pH 1.2), indicating rapid drug release by intact NA and NA-LDH. In contrast, in the coated sample, Eudragit^®^ S100 hinders LDH degradation by protonating its surface −OH groups under acidic conditions. Thus, in intestinal fluid, intact NA and NA-LDH exhibit the immediate release of NA, while the drug release is delayed in Eudragit^®^ S100-coated NA-LDH samples. Eudragit^®^ S100, an anionic polymer, replaces the encapsulated molecule (NA) efficiently by an ion-exchange reaction, facilitating the expansion of the LDH interlayer-space in neutral pH. Thus, it enables the construction of a controllable DDS, slowing the release of NA, alleviating its undesirable side effects, and enhancing its bioavailability. Starch [27] and alginate [61] pH-dependent coating systems have also been reported.

Carboxymethyl starch (CMS), an anionic water-soluble derivative of starch, has been utilized as a specific MTX intestinal-delivery agent in colon cancer therapy [27]. In the MTX delivery system, MTX intercalated in ZnAl-LDH is coated with CMS to protect the LDHs from gastric fluids. As shown in Figure 4b, in-vitro release and swelling tests in pH 1.2, 6.8, and 7.4 (colonic fluid) have been used to analyze the pH-sensitive oral drug delivery of the designed system; ZnAl-LDH/MTX (without CMS) and CMS@ZnAl-LDH/MTX have been used to analyze the effect of CMS. During experimentation, the sample without CMS dissolves completely under gastric conditions, whereas CMS-coated samples show controlled MTX release in intestinal and colonic-fluid environments (71.02% release within 8 h). This system, enabling the controlled delivery and release of MTX, exhibits high potential for application in colon-specific therapies involving oral administration. The drug-release performance and target-specific binding of these coated systems can be modulated using different ligands (for example, galactose or lactose is used for liver-tumor therapy [53,60], and dextran sulfate is used for actively identifying specific macrophage scavenger receptors [58]).

### 3.2. Cell Viability Test (In-Vitro)

The limitations of anticancer drug delivery include nonspecific cellular uptake, low absorption rates, variables in drug concentration, overcoming drug resistance, and others. LDHs, with low affinity to negatively charged cell membranes, exhibit high potential as cellular delivery nanocarriers for anionic anticancer drugs, including MTX [62] and 5-fluorouracil (5-FU) [63].

As shown in Figure 4c, Oh et al. have reported an intercalative anionic-polymer system using polyacrylic acid (PAA) for the cationic anticancer drug doxorubicin (DOX) [57]. DOX, one of the most widely administered commercial anticancer drugs, is used to treat different types of cancer, including bladder cancer, breast cancer, sarcoma, and others. Despite the high rate of cellular uptake and efficacy of DOX, an appropriate drug carrier is required to reduce its side effects (including cardiomyopathy, myelosuppression, and multi-drug resistance). DOX-PAA-LDH hybrids continuously release drugs to endosomal cells via endocytosis and exocytosis, as indicated by bio-transmission electron and confocal microscopic images. Both DOX and DOX-PAA-LDH show dose-dependent cell viability with two cancer-cell lines A549 and MG-63. Moreover, DOX-PAA-LDH exhibits a higher anticancer activity than DOX (at all DOX concentrations) without any delivery-agent cytotoxicity, indicating that the strategy reduces intracellular drug accumulation with low cytotoxicity. Moreover, this study reports a ten-times lower IC50 drug concentration than that reported in a previous publication [64].

Ghorbani’s group have reported a similar in-vitro cell study analyzing the biological activity of DOX against the liver-cancer cell line HepG2, as shown in Figure 4d [60], using hydroxyapatite (Ca_10_(PO_4_)_6_(OH)_2_, HAp), a cancer drug-delivery molecule, as a unique surface modifying agent for cancer treatment. HAp incorporation increases the LDH surface area and pore size, facilitating a higher DOX loading than that of unmodified LDH. To fabricate DOX/lactose@LDH-HAp nanocomposites, after the synthesis of HAp-modified LDH, DOX and the targeting ligand, lactose, are adsorbed on the surface and pores of HAp-LDH. The pH-responsive drug-release pattern of DOX/lactose@LDH-HAp nanocomposites indicates a higher rate of drug release at pH 5 (simulating the intracellular tumor environment) than at pH 7.4 (simulating normal tissue) under 37 °C for 72 h, possibly due to the presence of LDH. Additionally, the in-vitro cytotoxicity effects of Zn/Al/Ca-LDH, LDH-HAp, DOX, DOX@LDH-HAp, and DOX/lactose@LDH-HAp in the concentration range of 20–100 μg/mL against HepG2 cells have been analyzed using the MTT assay. The carrier materials Zn/Al/Ca-LDH and LDH-HAp exhibit no toxicity with HepG2, confirming their biocompatibility. Moreover, both DOX@LDH-HAp and DOX/lactose@LDH-HAp show higher cytotoxicity against HepG2 cells than the control drug, with lactose modification, which maximized the targeting ability of DOX. Therefore, these formulations enhance the cancer-treatment activity of DOX.

### 3.3. In-Vivo Study

Ideal nanomedicine made of LDH, in particular, must meet physicochemical and biological parameters for administration via the blood system during in-vivo trials. The physicochemical requirements primarily include size, aggregation status, and colloidal stability, while the biological requirements include the capacity to target specific areas such as disease cells and tissues, selective biological responsiveness, and low toxicity [65].

Oh et al. generated LDH nanoparticles intercalated with MTX using the ion exchange approach to address the safety and stability of MTX-LDH in blood vessel settings [66]. Before the in-vivo study, colloidal characteristics were examined by measuring the zeta potential and hydrodynamic diameter of MTX-LDH under simulated plasma conditions. At high doses, MTX is known to reduce the number of red blood cells significantly, resulting in hemolytic anemia. However, the suggested MTX-LDH exhibited almost no toxicity in blood cells even at high concentrations of 7.9 × 10^6^ and 7.6 × 10^6^ /µL, respectively, by confirming the degree of hemolysis and red blood cell number counting assay after 0.5 h and 6 h of intravenous injection to a Balb/c mouse in Figure 5a,b.

Although the nanohybrid system of LDH possesses exceptional benefits that appear very promising for medicinal applications, several obstacles still exist [67]. Owing to the positive charge of LDH, nanostructures can interact with negatively charged proteins in our biological environment and are, therefore, susceptible to surface charge loss and aggregation. The colloidal stability of LDH nanoparticles is crucial because particle aggregation in the medium can lead to blood vessel clotting, severe adverse effects, and animal mortality.

Bovine serum albumin (BSA) coating strategy on LDH nanoparticles for injection has been developed to improve dispersibility, delay the blood circulation time, and reduce the nonspecific cellular uptake [68,69,70]. An acid-responsive BSA−DOX prodrug and indocyanine green (ICG)-intercalated Cu-doped LDH NPs, ICG/Cu-LDH(@BSA-DOX) nanohybrid system, is an example of applying BSA as a stabilizing agent in serum during blood circulation, resulting in lowered rapid clearance and enhanced tumor accumulation rate in-vivo [71]. ICG can generate heating and produce reactive oxygen species (ROS) to achieve simultaneous photothermal and photodynamic therapy (PTT/PDT) for the treatment of cancer by near-infrared (NIR) irradiation. Cell membrane permeability enhances and promotes the accumulation of drugs by cancer cells through this reaction, thus improving cancer treatment effectiveness.

Figure 6a shows the results of a similar study by Zhu’s group, which describes a novel LDH-based multifunctional nanoplatform for co-delivery of anticancer medication, DOX and ICG as a synergistic chemotherapy, in addition to PTT [72]. Notably, arginine-tryptophan-(D-arginine)-asparagine-arginine (B3int) can target cancer cells and enhance the entry of nanomaterials and B3int into the tumor zone. Utilizing surface adsorption to load both molecules, 808 nm-irradiated DOX-ICG@LDH-poly ethylene glycol (PEG)-B3int demonstrated superior tumor growth inhibition compared to saline or free DOX in a mouse model injected with murine melanoma cells. The high biocompatibility and safety of LDHs as a nanocarrier was demonstrated by monitoring body weight and the percentage of mice that survived the observation period (Figure 6b–e).

Recently, an in-vivo study was conducted using a combination of chemotherapy and phototherapy, using anticancer medications such as 5-FU, albumin-bound paclitaxel [73], and the anti-malaria agent dihydroartemisinin (DHA) [74]. Table 1 lists the drug delivery reports from 2019 to 2023, including the ones specified.

## 4. Design of Functional LDH Nanohybrids for Imaging

In addition to therapeutic applications, LDHs have been analyzed for numerous novel bio-imaging applications. The interaction routes of nanoparticles, including the interaction of LDHs with biological surroundings, could be used to understand their treatment mechanisms; thus, it is vital to develop highly sensitive, trustworthy, and robust labeling methods for reactant tracking [77].

This section focuses on recent (published from 2019 to 2023) in-vitro and in-vivo studies [49] on fluorescence imaging [78], magnetic resonance imaging (MRI) [79], and positron emission tomography (PET) [28] without any therapeutic molecules (Figure 7). Functional LDH nanohybrids for bio-imaging applications are of three main types: First, an intercalation type, in which an MRI-active or fluorescent substance, such as FITC [32,80], Gd(III) complex [81], or diethylenetriamine pentaacetic acid (Gd-DTPA) [82,83], is intercalated into the LDH interlayer spaces. Second, a substitutive type, in which a metal ion (such as MRI-active Gd^3+^ and Mn^2+^ ions or PET-active ^64^Cu^2+^ [68]) is doped into the metal hydroxide layer. Third, a surface modification type, which utilizes surface adsorption for imaging applications [79].

### 4.1. Fluorescence Imaging Techniques

To track the pathway of a particle in a biological system, the particle is labeled with a fluorophore. Fluorescence imaging techniques have attracted immense attention for bio-imaging because of their ability to acquire high spatial resolution images with good sensitivity. Moreover, their short acquisition time facilitates real-time monitoring. With the rapid development of nanotechnology, numerous studies have analyzed bio-imaging at the subcellular or molecular level [80]. Additionally, the applicability of diverse fluorescent dyes, including fluorescein [84,85], FITC [86,87,88], 8-aminonaphthelene-1,3,6-trisulfonic acid (ANTS) [89], and 8-aminopyrene-1,3,6-trisulfonic acid (APTS), for the biological tracking of LDH nanoparticles has been investigated [77].

Xu and Sun’s group have reported an enhancement in the tumor accumulation and cellular uptake of LDH nanoparticles by complexing them with the negatively charged polymer, dimethylmaleic acid (DMMA)/poly(2-aminoethyl methacrylate hydrochloride (PAMA)), due to electrostatic interactions [78]. To track the pH-dependent cellular accumulation of particles, the fluorescent dye, FITC, is intercalated into the interlayer space of LDH. As shown in Figure 8a, confocal laser scanning microscopy (CLSM) images indicate a significant intensity difference between FITC signals at pH 7.4 and 6.8 on treating FITC/Cu-LDH@PAMA/DMMA NPs for 4 h with normal (RAW 264.7) and cancer (B16F0) cells. The authors have suggested that the synthesized nanoparticles are internalized to a lesser extent under physiological conditions (pH 7.4) due to electrostatic repulsion between the polymer-coated LDH molecules and negatively charged cell membranes. However, under slightly more acidic conditions, at pH 6.8 (tumor microenvironment), the pH-dependent PAMA/DMMA polymer hydrolyzes, forming positively charged pristine Cu-LDH NPs which adhere to the negatively charged cytomembrane, thereby enhancing the cellular uptake of nanoparticles. Additionally, Cao’s group has reported the ex-vivo fluorescence imaging of rabbit ocular tissue with FITC-loaded samples [88].

However, the biomedical applications of in-vivo fluorescence imaging are limited due to the short tissue penetration depth, light scattering, low spatial resolution due to autofluorescence and adsorption by adjacent tissues, water, and lipids as observed in biological systems [91,92].

### 4.2. MRI Techniques

Among the numerous imaging methods, MRI [6,82,93,94,95], with a noninvasive nature and excellent penetration depth, is particularly suitable for soft-tissue diagnosis in cancer and stroke treatment; it makes anatomical information visible, providing 3D tomographic images of whole tissues and lesions. MRI is typically operated under the magnetic moments generated from protons in the water interacting with cell membranes, providing information regarding tissues under large magnetic fields of high magnitude. Signals from the process are transmitted under radio frequencies to produce images. The MRI resolution can be significantly improved by using contrast agents to shorten the T_1_ (longitudinal relaxation time) or T_2_ (transverse relaxation time) of water. Therefore, although two signals exist due to the relaxation process, the T_1_ signal in T_1_-weighted imaging appears brighter due to contrast effects, whereas the T_2_ signal in T_2_-weighted imaging appears dark when T_2_ contrast agents are used.

Paramagnetic cations, such as manganese [96,97] and gadolinium [98,99], and gadolinium chelates such as Gd(DTPA) [82,100] have been used as T_1_ contrast agents, and the contrast in T_2_ signals has been enhanced by superparamagnetic iron oxide [101,102].

However, endogenous variations interrupt the original MRI signals, generating uncertainty with single-mode contrasting agents. Therefore, Gu’s group has designed a dual-function LDH (T_1_/T_2_ dual-mode LDH) by combining T_1_ and T_2_ contrast agents with pH response, as shown in Figure 8b [79]. They have reported manganese-based magnetic LDH nanoplates labeled MgMnAl-LDH@iron oxide (IO) NPs, fabricated by conjugating a positive layer of MgMnAl-LDH with negatively charged IO particles. The T_1_, T_2_ magnetic resonance images of MCF-7-tumor-bearing mice, shown in the lower part of Figure 8b, indicate the pre/post T_1_ and T_2_ signals after injecting the mice with BSA/MgMnAl-LDH@IO NPs coated with BSA for an in-vivo study. As shown in the image, 24 h after injection, the T_1_ image increases in intensity from 458.8 ± 15.9 to 1091.6 ± 66.4 (brighter image), whereas the T_2_ signal becomes darker (with an intensity reduction from 873.2 ± 33.3 to 32.6 ± 10.1). This in-vivo study indicates a concurrent enhancement in both the modes of magnetic bio-imaging by MgMnAl-LDH@IO NPs, indicating a high potential for accurate tumor diagnosis.

Additionally, an MRI method utilizing ^19^F (instead of ^1^H) has attracted immense attention due to interactions of ^19^F with cancer cells. Typically, most clinical MRIs involve ^1^H detection; however, high background signals due to water and intrinsic contrast signals, including those due to blood clots, limit the imaging capability of this technique [103]. Although the ^19^F molecule exhibits high potential for MRI due to fluorine quantification and ^19^F MRI-signal variations with microenvironment changes, it is challenging to synthesize high-emission fluorinated systems with high resolution for cancer cells. Xu and Whittaker’s group have reported a novel manganese-LDH nanohybrid integrated with a fluorinated polymer, perfluoropolyether, to form a surface-organizing pH-responsive on/off system, as shown in Figure 8c [90]. At pH 7.4, the off-system operates; as ^19^F nuclei approach the surface of LDH, Mn(II) atoms cause a paramagnetic relaxation in the ^19^F atoms. This significantly reduces the nuclear magnetic resonance (NMR) T_2_ paramagnetic relaxation time of the fluorinated polymer, and no signal is detected. In contrast, the on-system functions for pH values below 6.8; Mn(II) exists as Mn^2+^, formed due to the destruction of LDH under acidic conditions. This transforms the ^19^F T_2_ signal to its original intensity, causing intense ^19^F NMR and MRI signals. Figure 8c B shows that in-vivo ^19^F MRI scans show breast cancer 24 h after injecting BSA-coated Mn-LDH@PFPE nanoparticles into an MDA-MB-468 subcutaneous murine tumor model. Unlike the normal organs (the liver, lung, and kidney), strong ^19^F MRI signals are observed at the tumor sites. Moreover, alternations in the T_1_-weighted ^1^H and ^19^F MRI signal intensities in the tumor region have been checked at numerous time points between 0 to 96 h. Until 24 h post-injection, the intensity of the ^19^F signal increases considerably in comparison with the T_1_-weighted ^1^H signal, possibly because Mn^2+^ atoms located nearby affect the ^19^F nuclei under the tumor microenvironment to a lesser extent; after 24 h, the signal intensity increases slightly due to the gradual dissolution of LDH.

### 4.3. Positron Emission Tomography

PET, a molecule-based noninvasive diagnostic imaging method with excellent sensitivity, is particularly suitable for obtaining functional information about slight changes in disease biomarkers in in-vivo systems without any tissue penetration limit [28,104,105].

Typically, PET is conducted after intravenously injecting a small quantity of radioactive tracers, such as the positron-emitting isotopes ^11^C [106,107], ^18^F [108,109], ^68^Ga [110], and ^13^N [106] that are commonly used for in-vivo analysis, into the system. Numerous studies have combined these radioisotopes with metal oxides, dendrimers, chelators, liposomes, and layered compounds [111,112,113]. Typically, ^18^F-fluorodeoxyglucose (FDG), ^11^C-methionine (MET), and ^13^N-ammoniasome (NH_3_), with one -OH replaced by an isotope, are used for PET. However, systems with ^18^F and ^68^Ga have short half-lives, making them unsuitable for in-vivo studies; moreover, the limited evaluation of brain tumors in FDG-PEG and MET systems could generate high-intensity signals for non-tumor lesions.

To overcome these limitations, Choy’s group has reported the use of ^64^Cu as a tracing agent; it is suitable for PET imaging owing to its relatively long half-life (12.7 h), decay by multiple pathways including β^+^ energy emission (18%), and high production yield and purity [28]. As illustrated in Figure 9a, chemically stable ^64^Cu-labeled quintinite nanoplate (QT-NP) are synthesized by coprecipitation and subsequent hydrothermal treatment (to immobilize ^64^Cu^2+^ ion into the octahedral sites of quintinite in a ratio that is different from that in naturally existing LDHs), followed by coating with BSA (for enhancing their stability during blood-stream circulation). Various techniques, including powder XRD and SEM, have been used to analyze their characteristics (including particle size, surface charge, and stability). Moreover, their PET imaging quality has been investigated by in-vivo experiments using an MDA-MB-231 xenograft mouse model; at a specific time after sample injection, coronal and significant transverse PET images are produced by the gamma rays generated by ^64^Cu isotopes (511k eV) (Figure 9b). BSA-coated samples (^64^Cu-labeled QT-NPs/BSA) generate significantly brighter images than uncoated samples, confirming that BSA prolongs the nanohybrid circulation time, facilitating the tumor-cell sample uptake (24 h after injection, the tumor cells exhibit a 2-fold higher sample content). Quantitative region-of-interest (ROI) analysis has been used to construct time activity curves (that depend on tumors, muscles, liver, and blood) to analyze the passive targeting effect of the QT-NPs; they indicate an instant and continuous uptake of the nanohybrid into tumor tissues (Figure 9c). In accordance with the process of enhanced permeability and retention (EPR) and BSA effects, the tumor-to-liver ratio (%) of ^64^Cu-labeled QT-NPs/BSA nanoparticles (21.86 ± 5.05%) is 1.7-fold higher than that of ^64^Cu-labeled QT-NPs (12.67 ± 6.38%), 48 h after injection (Figure 9d), indicating that the former is internalized into targeted tumor tissues to a greater extent than the latter. LDH nanomaterials related to imaging applications without any drug (published from 2019 to 2023) are summarized in Table 2.

## 5. Design of Functional LDH Nanohybrids for Theranostics

Theranostics, formed by combining the words therapeutics and diagnostics, indicates the simultaneous diagnosis and treatment of diseases. Since the first report on LDH-based nanohybrids in 1999, numerous studies have aimed to engineer LDHs for biomedical applications, including drug delivery and imaging. Recent studies have reported LDH-incorporated multifunctional nanohybrids combining therapeutic drug molecules and other imaging moieties as effective platforms for in-vivo imaging and therapy [81,114,115,116,117,118,119,120]. A single LDH system can be used for chemotherapy, photo therapy, and gene therapy; thus, one nanohybrid can exhibit multiple therapeutic activities [119,121,122,123].

### Evaluation Based on In-Vivo Study

Although in-vivo fluorescence imaging has a limited penetration depth, fluorescent nanosystems for drug monitoring are being developed in the ocular field [124]. According to a study of flurbiprofen (FB) for ocular delivery as a non-steroidal anti-inflammatory drug, a hyaluronic acid (HA)-coated FB intercalated-LDH ophthalmic DDS (HA-FB-LDH) [125] has been designed. In-vitro cumulative release levels of FB-LDH and HA-FB-LDH within 12 h were 92.99 ± 0.37% and 74.82 ± 0.29%, respectively, indicating a sustained release. Even after 30 min of administration, the fluorescence of the FB-LDH and HA-FB-LDH groups was still quite bright, indicating a residual of the HA-FB-LDH. The fluorescence imaging approach was employed to detect fluorescence signals in rabbits’ eyes for the ocular surface retention of the FB eye drops, each formulation with FITC labeling. LDH materials were included. FITC-LDH and HA-FITC-LDH could both lengthen drug precorneal residence duration, indicating that LDH adhesion slows the clearance rate of tears on the ocular surface, improving flurbiprofen ocular bioavailability.

Zhou’s group has utilized the excessive reactive oxygen species (ROS) generated by ischemic stroke reperfusion for its treatment via an atorvastatin-ferritin Gd-LDH nanohybrid (AFGd-LDH) [81]. With high ROS-capture efficiency, this system can move through the blood-brain barrier (BBB) and enable in-vivo visualization due to its MRI imaging ability. The BBB is a type of protective layer in the central nervous system (CNS) that shields the brain from bloodborne pathogens. However, it also restricts therapeutic efficacy and presents a challenge in developing drugs that target the brain for brain cancer and neurodegenerative diseases [126]. MgAlGd-LDH is synthesized by the coprecipitation of Gd^3+^ and Al^3+^, followed by the intercalation of the anti-inflammatory drug anti-lipid peroxidative atorvastatin (ATO) and ferritin (FTH) (which enables the particle to cross the BBB via endocytosis by binding to the transferrin receptor 1 (TfR1) of endothelial cells between the layer). AFGd-LDH exhibits a higher ROS-capture ability than CeO_2_, in the form of a clinical neuroprotective drug edaravone (PMP), indicating a combined influence of AF-LDH and ATO on the activity of the nanohybrid. Additionally, AFGd-LDH acts as an effective MRI contrast agent for brain imaging; it exhibits an excellent in-vivo MRI performance with reducing neurons apoptosis and oxidative damage in the brain cortex (tMCAO) mice model, 3 h after surgery, with brighter signals for infarction lesions. Ta’s group has reported a similar study involving ROS overproduction and MRI scans using CeO_2_-Fe_3_O_4_@LDH nanocomposites [127].

Similar to its combination with MRI, drug delivery has also been synergized with nuclear imaging techniques such as single photon emission computed tomography (SPECT), PET, and tracer-based medical imaging modalities (Figure 10). Whereas SPECT uses tracers that directly emit single high-energy photons, PET is based on detecting photon pairs generated by annihilating positrons and electrons [128].

Oh’s group has developed a tumor-targeting theranostics nanomedicine that can be used for MTX delivery as well as SPECT imaging (by intercalating the radioisotope and anticancer agent into LDH) [29]. Figure 10a shows the strategy of this biomedical application. The system is prepared by incorporating an anticancer drug, MTX, into the LDH interlayer space and a radioisotope, Co-57, into the interlayer MTX moiety; thus, Co^2+^ is coordinated to the electron pairs of MTX (on the abundant base sites in the pteridine part). A time-dependent release test has been used to evaluate the release of Co-57 from Co-57@MTX-LDH; the fraction of release in 48 h is ~0% and 12% in PBS and human serum, respectively, due to the insolubility of LDH in physiological solutions. In-vitro and in-vivo anticancer-activity analyses of MTX-LDH indicate low cell viability in the mouse colon carcinoma cell line (CT-26) at administration doses in the range of 5–500 μg/mL; with 5 µg/mL of MTX-LDH, 32.5 ± 9.17% cell viability is observed (Figure 10b). The morphology of CT-26 cells confirms that the rate of internalized MTX-LDH gradually increases on increasing the dose of administration, enhancing drug delivery to the cell (Figure 10c). As shown in Figure 10d–f, considering the injection of Co-57@MTX-LDH into a xenografted mouse model with a CT-26 tumor, the in-vivo SPECT signal in the tumor tissue appears an hour after injection, it is maximized after 3 h, and subsequently reduces (6 h post-injection). Table 3 organizes the above-mentioned study, combined with drug administration and imaging application.

## 6. Conclusions

This review provides a summary of the relevant literature on LDHs, enabling researchers to understand the utilities and challenges of the diverse functional systems reported and suggest a direction for future analysis. Recently developed LDH nanomaterials with biomedical applications (such as drug delivery and bio-imaging) and the results of their in-vitro and in-vivo studies after drug incorporation are listed in Table 1, Table 2 and Table 3 to guide the design of new nanotechnology with LDHs for therapy and diagnosis. As shown in Figure 11, the number of citations related to LDHs with biomedical applications has rapidly increased with using such systems as tools for technological advancement and progress in biomedicine.

LDHs have numerous advantages as a DDS, including biodegradability, biocompatibility, low cytotoxicity, high loading efficiency for anionic molecules, and pH-dependent drug release. Additionally, incorporating contrast agents into LDHs enhances their imaging properties, allowing for better visualization of biological structures. Real-time theranostics, which combine imaging and therapy, hold great promise for early disease detection. The design and development of highly stable LDH drug-delivery agents can be achieved by modifying their surfaces with polymers, biomolecules, and other nanoparticles. If they can circulate in physiological media without aggregating for long periods, they could significantly contribute to biomedical research and real-time theranostics.

In the future, LDH-supported drug/imaging agent systems are expected to become more useful in clinical applications as effective drug-delivery and noninvasive bio-imaging techniques continue to improve. 

## Figures and Tables

**Figure 1 nanomaterials-13-01102-f001:**
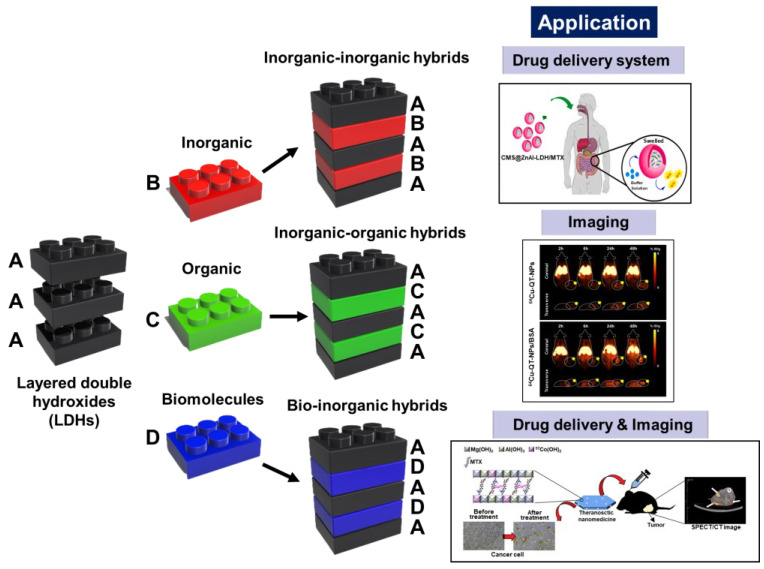
Schematic illustration of the formation of specific-application nanohybrids through the 2D lattice engineering of 2D host building blocks (layered double hydroxides). The application images are reprinted with permission from Refs. [27,28,29], Copyright 2022, Elsevier; Copyright 2022, the Royal Society of Chemistry; and Copyright 2020, Elsevier, respectively.

**Figure 2 nanomaterials-13-01102-f002:**
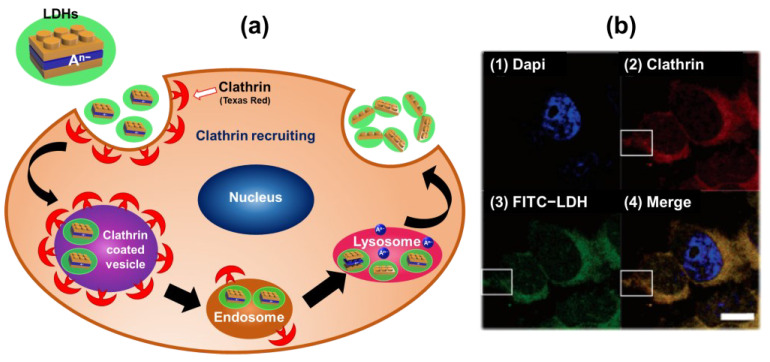
Uptake pathway of layered double hydroxides (LDHs) into cells. (**a**) A representative process of clathrin-mediated endocytosis (**b**) Confocal microscopy, which includes the colocalization of fluorescein 5′-isothiocyanate (FITC)-LDH and clathrin in MNNG/HOS cells (involving the localization of (1) the nucleus, (2) clathrin, and (3) FITC-LDH in MNNG/HOS cells, as shown in (4) the merged image). Cells were incubated with FITC-LDH for 2 h, treated with clathrin antibodies, and stained by Texas Red and 4′,6-diamidino-2-phenylindole. Scale bar = 10 µm. (Reprinted with permission from Ref. [31], Copyright 2006, the American Chemical Society.).

**Figure 3 nanomaterials-13-01102-f003:**
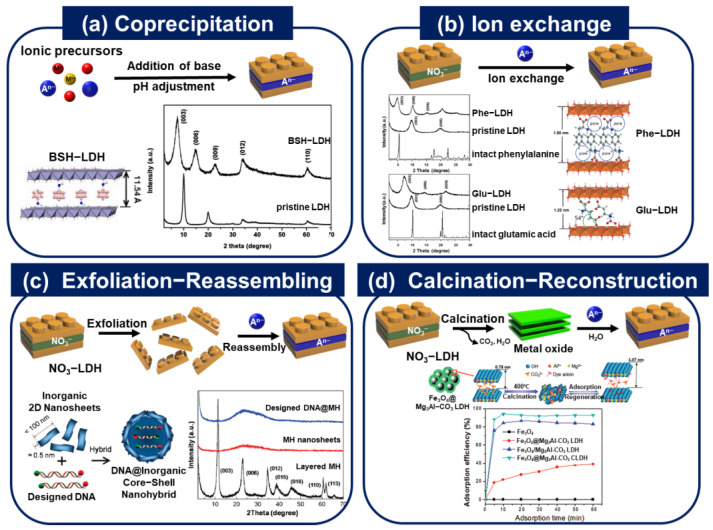
Several methods to prepare lattice-engineered layered double hydroxide (LDH) nanohybrids. (**a**) Coprecipitation: structure of BSH-LDH, powder X-ray diffraction (XRD) patterns for pristine LDH and BSH-LDH (reprinted with permission from Ref. [36], Copyright 2017, John Wiley & Sons Inc.). (**b**) Ion exchange: powder XRD patterns of intact phenylalanine, pristine LDH, Phe-LDH and intact glutamic acid, pristine LDH, Glu-LDH, and structure of prepared Phe-LDH and Glu-LDH (reprinted with permission from Ref. [3], Copyright 2015, John Wiley & Sons Inc.). (**c**) Exfoliation−Reassembling: scheme for designed DNA@inorganic core-shell nanohybrid and powder XRD patterns of layered MH, MH nanosheets, and designed DNA@MH (reprinted with permission from Ref. [41], Copyright 2010, the American Chemical Society). (**d**) Calcination−Reconstruction: scheme of the phase evolution of the Mg_3_Al–CO_3_ LDH shell during calcination, dye adsorption, and regeneration and comparison of the activity of Fe_3_O_4_, Fe_3_O_4_@Mg_3_Al–CO_3_ LDH, core-shell Fe_3_O_4_@Mg_3_Al–CO_3_ CLDH, and hybrid Fe_3_O_4_/Mg_3_Al–CO_3_ CLDH (reprinted with permission from Ref. [35], Copyright 2015, John Wiley & Sons Inc.).

**Figure 4 nanomaterials-13-01102-f004:**
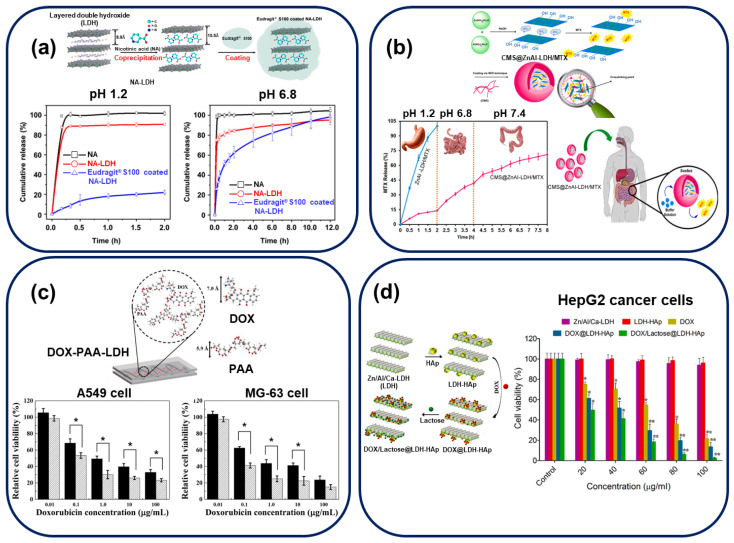
Drug delivery systems analyzing the in-vitro drug release and therapeutic actions of LDH. All experimental results are included in the synthetic illustration of the reported LDH-modification strategy. (**a**) In-vitro release profiles of NA from intact NA (─□─), NA-LDH (**─○─**), and Eudragit^®^ S100-coated NA-LDH (**─△─**) in pH 1.2 and 6.8 media. (Reprinted with permission from Ref. [5]) (**b**) MTX-release profiles of free and CMS-coated ZnAl-LDH/MTX under in-vitro simulated GIT conditions and schematic illustration of the oral delivery process of CMS@ZnAl-LDH/MTX. (Reprinted with permission from Ref. [27]. Copyright 2022, Elsevier) (**c**) Cytotoxicity of DOX (black bar) and DOX-PAA-LDH (patterned bar) against A549 and MG-63 cells. Asterisks indicate statistical differences with confidence intervals of 95% calculated by the student’s *t*-test. (Reprinted with permission from Ref. [57]. Copyright 2021, Elsevier) (**d**) In-vitro cytotoxicity assay curves of Zn/Al/Ca-LDH, LDH-HAp, DOX, DOX@LDH-HAp, and DOX/lactose@LDH-HAp with HepG2 cancer cells after 48-h incubation. The results are calculated as mean ± standard deviation (*n* = 3), * *p* < 0.05, ** *p* < 0.01. (Reprinted with permission from Ref. [60]. Copyright 2022, Elsevier).

**Figure 5 nanomaterials-13-01102-f005:**
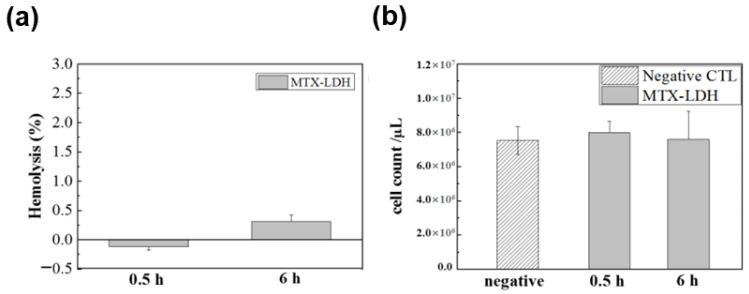
In-vivo (**a**) hemolysis and (**b**) red blood cell number counting assay results of MTX-LDH after 0.5 h and 6 h of injection to a Balb/c mouse. The MTX-LDH suspension (10 mg/mL) was injected (100 µL) through the tail vein, and the blood was analyzed at each time point. (Reprinted with permission from Ref. [66]).

**Figure 6 nanomaterials-13-01102-f006:**
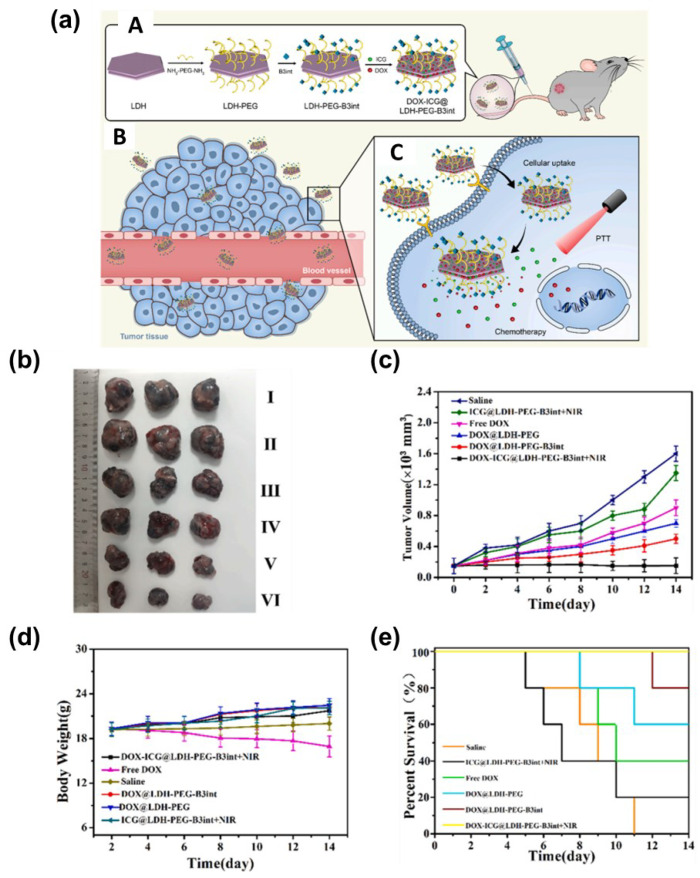
(**a**) A. Schematic illustration of the synthesis of DOX-ICG@LDH-PEG-B3int NPs. B. Cartoon representation of how DOX-ICG@LDH-PEG-B3int NPs remain in the tumor site through the targeting effect. C. The release of DOX and ICG from the DOX-ICG@LDH-PEG-B3int NPs in the tumor microenvironment. (**b**) The images of tumors isolated after treatment for 14 d (I: Saline; II: ICG@LDH-PEG-B3int+NIR; III: Free DOX; IV: DOX@LDH-PEG; V: DOX@LDH-PEG-B3int; VI: DOX-ICG@LDH-PEG-B3int+NIR). (**c**) B16 tumor growth curves of different groups after intravenous injection of the formulations. (**d**) Body weight changes over the 14 d of the experiments. (**e**) Kaplan–Meier survival curves (*n* = 3). (Reprinted with permission from Ref. [72], Copyright 2021, Elsevier).

**Figure 7 nanomaterials-13-01102-f007:**
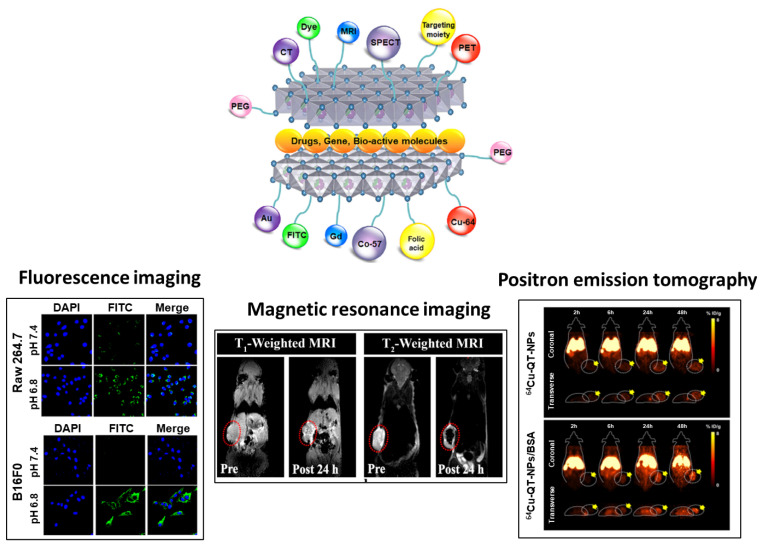
Layered double hydroxide (LDH) nanohybrids for imaging applications (reprinted with permission from Ref. [49], Copyright 2020, John Wiley & Sons Inc.), including fluorescence (reprinted with permission from Ref. [78], Copyright 2021, the American Chemical Society), magnetic resonance imaging (reprinted with permission from Ref. [79], Copyright 2019, the American Chemical Society), and positron emission tomography imaging (reprinted with permission from Ref. [28], Copyright 2022, the Royal Society of Chemistry).

**Figure 8 nanomaterials-13-01102-f008:**
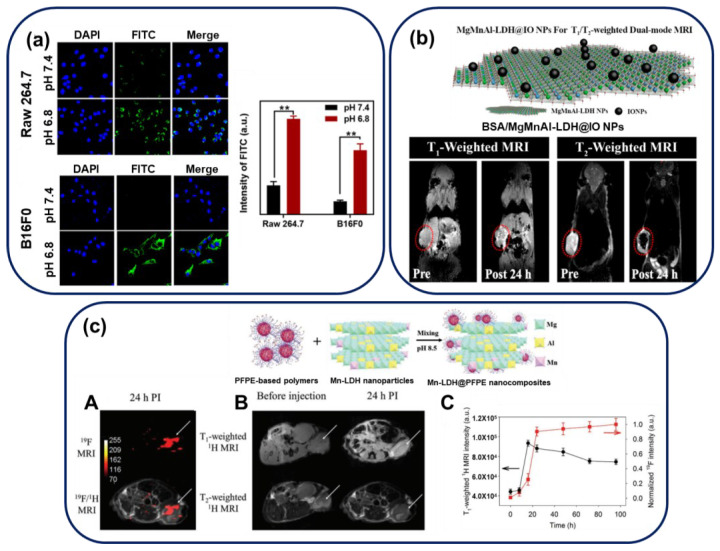
(**a**) Uptake of Cu-LDH@PAMA/DMMA by B16F0 and RAW 264.7 cells. CLSM images of RAW 264.7 cells and B16F0 cancer cells and their mean fluorescence intensities via flow cytometric analysis after treatment with Cu-LDH@PAMA/DMMA in media with pH 7.4 and 6.8 at 37 °C for 4 h; ** *p* < 0.01. (Reprinted with permission from Ref. [78], Copyright 2021, the American Chemical Society) (**b**) T_1_- and T_2_-weighted MRI of breast-tumor-bearing mice before and after intratumoral injections of BSA/MgMnAl-LDH@IO NPs. (Reprinted with permission from Ref. [79], Copyright 2019, the American Chemical Society) (**c**) Schematic illustration of the synthetic procedure of Mn-LDH@PFPE nanoparticles and analyzing their utility for the specific imaging of cancer using a mouse subcutaneous MDA-MB-468 tumor model. For each mouse, 200 μL of an Mn-LDH@PFPE PBS solution (1.8 mmol fluorine per kg = 400 mg per kg or 8 mg per mouse) was injected intravenously through the tail vein (*n* = 3). A. ^19^F MRI images (at 9.4 T) of the tumor-bearing mice before and after (24 h PI) intravenous injection of Mn-LDH@PFPE nanoparticles. B. ^1^H T_1_- and T_2_-weighted MRI images, 24 h after injecting Mn-LDH@PFPE nanoparticles. Acquisition times for ^1^H T_1_- and T_2_-weighted MRIs are 1 min 42 s and 1 min 19 s, respectively. The white arrows in (**c**) A and B indicate tumor sites. C. Changes in T_1_-weighted ^1^H and ^19^F MRI signal intensities in the tumor region after injecting Mn-LDH@PFPE nanoparticles. Data in C indicate mean ± standard deviation (SD) values. (Reprinted with permission from Ref. [90], Copyright 2019, John Wiley & Sons Inc.).

**Figure 9 nanomaterials-13-01102-f009:**
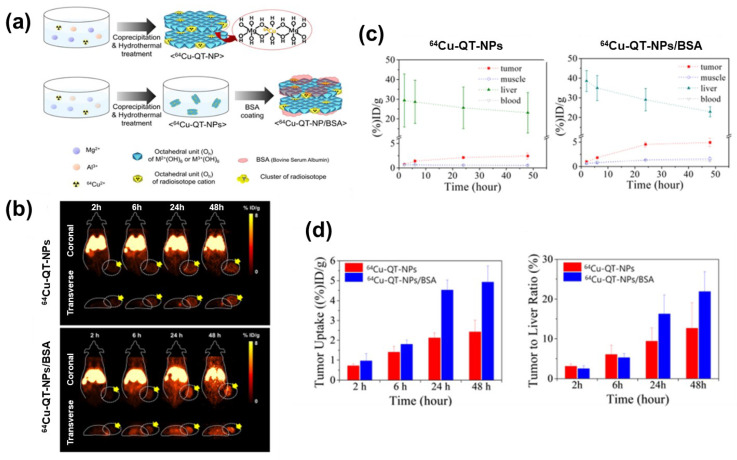
(**a**) Structural schemes for isomorphically substituted ^64^Cu in the QT lattice of ^64^Cu-QT-NPs and BSA-coated ^64^Cu-QT-NP/BSA, (**b**) In-vivo PET images of ^64^Cu-QT-NPs and ^64^Cu-QT-NPs/BSA in an MDA-MB-231 xenograft mouse model after intravenous injection. (Tumor region: white-dotted circles indicated with yellow arrows) (**c**) Time–activity curves and mean values ± SD (*n* = 3) for the tumor, muscle, liver, and blood cells for (left) ^64^Cu-QT-NPs and (right) ^64^Cu-QT-NPs/BSA. (**d**) The tumor-tissue uptake rate and tumor-to-liver ratio (%) of ^64^Cu-QT-NPs and ^64^Cu-QT-NPs/BSA in an MDA-MB-231 xenograft mouse model, 2, 6, 24, and 48 h after injection. (Reprinted with permission from Ref. [28], Copyright 2022, the Royal Society of Chemistry).

**Figure 10 nanomaterials-13-01102-f010:**
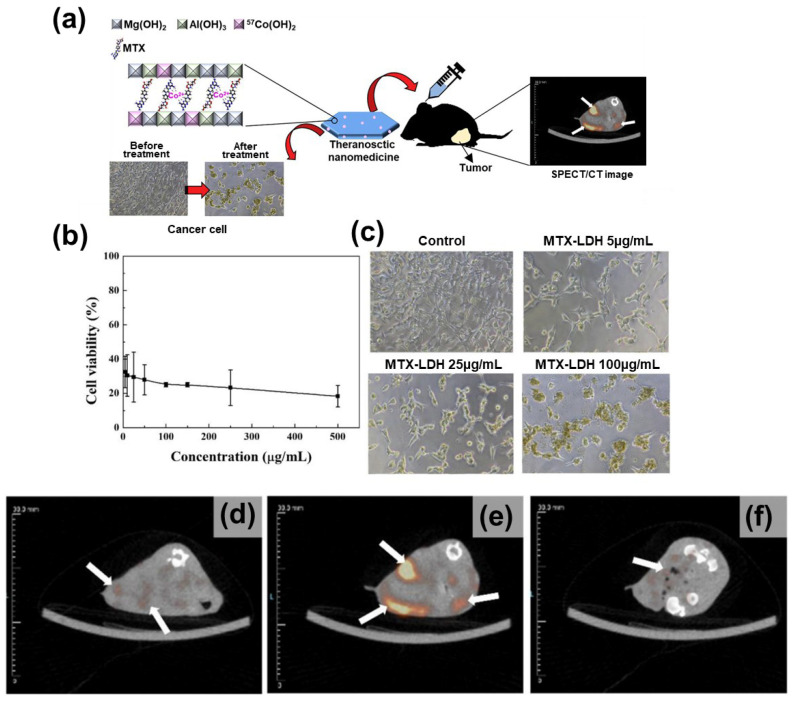
(**a**) Graphical abstract of the paper reporting Co-57 labeled MTX-LDH. (**b**) In-vitro cell viability test, 24 h after MTX-LDH injection. (**c**) Morphologies of CT-26 cells on varying the concentration of injected MTX-LDH. (**d**–**f**) SPECT/CT images of a tumor in the mouse model, (**d**) 1 h, (**e**) 3 h and (**f**) 6 h after the intravenous injection of Co-57@MTX-LDH. (Reprinted with permission from Ref. [29], Copyright 2020, Elsevier).

**Figure 11 nanomaterials-13-01102-f011:**
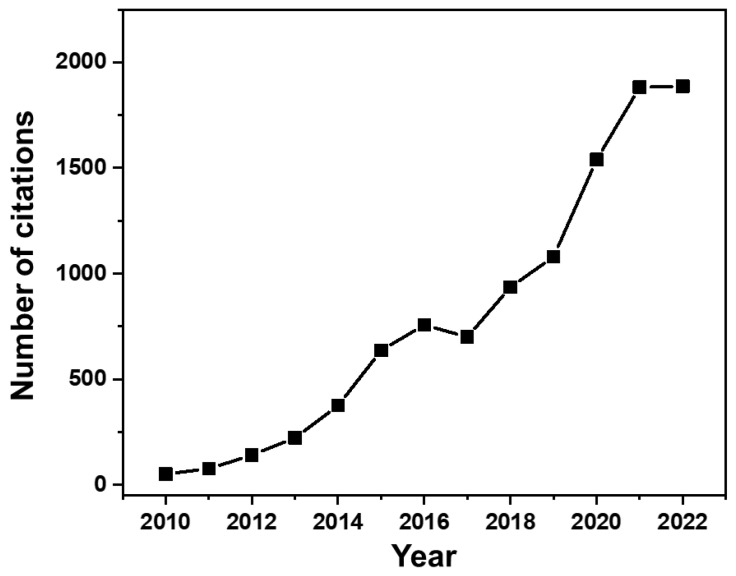
Number of citations annually on biomedical applications of layered double hydroxides. (Source: Web of Science).

**Table 1 nanomaterials-13-01102-t001:** Summary of LDH nanomaterials related to drug delivery applications reported from 2019 to 2023.

	Synthetic Approaches	Types of Biomedical Applications	Active Molecules (Therapeutic Agents or/and Contrasting Agents)	Additional Agent for Stability	Remarks	Ref.
ZnAl-LDH	Coprecipitation, Ion-exchange	Drug delivery	Intercalation of methotrexate (MTX)	Carboxymethyl starch (CMS) as a pH-sensitive polymer	CMS@ZnAl-LDH/MTX (23.7%), with biodegradability, is used for controlled drug delivery to the lower gastrointestinal tract. The carrier shows good biocompatibility with HT 29 cells	[27]
ZnAl-LDH	Coprecipitation	Drug delivery	Intercalation of nicotinic acid (NA)	Eudragit^®^ S100 for enteric coating	NA (26.1%)-LDH causes lesser side effects than NA, with better pharmacological effects, without undesirable toxicity	[5]
ZnAlCa-LDH	Coprecipitation	Drug delivery	Surface incorporation of doxorubicin (DOX)	Lactose as a specific-target ligand, hydroxyapatite for surface modification	DOX/lactose@LDH-HAp shows controlled and pH-dependent drug release, and its high cancer-cell suppression (tested with HepG2 cells) confirms the high potential for the targeted and controlled delivery of DOX	[60]
MgAl-LDH	Coprecipitation	Drug delivery	Intercalation of 5-fluorouracil (5-FU) and surface absorption of doxorubicin (DOX)	Carboxymethyl starch (CMS) as a pH-sensitive polymer	An acceptable formulation for oral co-drug delivery; it shows an acceptable controlled drug-release profile (~22% for DOX and 29% for 5-Fu), according to in-vitro analysis and a cell (Caco-2) viability test	[56]
MgAl-LDH	Coprecipitation-Hydrothermal	Drug delivery	Intercalation of curcumin (Cur)	Galactose (Gal) as a cancer-specific ligand	Gal-Cur (31.0%)/LDH is an excellent carrier system for the targeted delivery of Cur to HepG2 cells via an asialoglycoprotein (ASGP) receptor-mediated pathway with a high rate of apoptosis	[53]
MgAl-LDH	Coprecipitation	Drug delivery	Intercalation of indomethacin (INDO)	-	LDH-INDO (50%) enables a faster recovery than intact INDO from LPS-induced inflammatory processes, according to in-vivo assays on male Wistar rats	[55]
MgAl-LDH	Coprecipitation, Hydrothermal, Ion-exchange	Drug delivery	Intercalation of doxorubicin (DOX)	Polyacrylic acid (PAA) for surface modification	DOX-PAA-LDH shows anticancer activity; it efficiently delivers drugs to the nucleus, according to tests with two cancer cells, MG-63 and A549, analyzed by confocal microscopy and cellular assays	[57]
MgAl-LDH	Coprecipitation	Drug delivery	Intercalation of insulin	Alginate for preparing hydrogel, chitosan for improving stability	Chitosan-alginate-based insulin (~85%) LDHs are a promising platform for sustained oral insulin delivery under pH 6.8, according to tests on the chicken intestine	[61]
ZnAl-LDH	Coprecipitation	Drug delivery	Intercalation of captopril (CPL)	-	By intercalating an antihypertensive drug, CPL, into the interlayer spaces of zinc–aluminum-LDH, providing efficient controlled release formulation with less toxicity compared to pure CPL is achieved	[75]
MgAl-LDH	Coprecipitation, Hydrothermal, Ion-exchange	Drug delivery	Intercalation of pemetrexed (PMX)	Sericin (Seri) protein and glutaraldehyde (GTA) for stabilizing agent	Sustained release manner (99.23% after 30 h and 99.3% after 11 h at pH 7.4 and 4.8, respectively) with high cellular uptake performance for hydrophilic PMX is shown synthesizing 3-triethoxysilylpropan-1-amine (APTES)-ZnO quantum dots (QDs)/Seri@LDH2-PMX (GTA)	[76]
MgAl-LDH	Coprecipitation, Hydrothermal, Ion-exchange	Drug delivery	Intercalation of methotrexate (MTX)	-	In-vivo hemolysis assay after intravenous injection of MTX-LDH showed neither significant reduction in red blood cell number nor membrane damage without no change of morphology of red blood cells in mice, taking advantage of the cell hitchhiking property.	[66]
Cu-LDH	Isomorphic substitution	Drug delivery, phototherapy	Intercalation of indocyanine green (ICG), Surface incorporation of doxorubicin (DOX)	Bovine serum albumin (BSA) for stabilizing agent	ICG/Cu-LDH@BSA-DOX release DOX in an acid-triggered manner and eradicated the tumor tissues upon very low doses of therapeutic agents (0.175 mg/kg DOX, 0.5 mg/kg Cu, and 0.25 mg/kg ICG) upon 808 nm laser irradiation	[71]
MgAl-LDH	Hydrothermal	Drug delivery, phototherapy	Physical adsorption of doxorubicin (DOX) and indocyanine green (ICG),	H_2_N-PEG-NH_2_ as a linker of targeting agent, B3int, for improving biocompatibility and hydrophilicity	Synergistic photothermal therapy (PTT)/chemotherapy utilizing ICG and DOX coloaded LDH (LDH -PEG-B3int NPs) through pH-responsive and near-infrared (NIR)-triggered DOX release having anti-tumor activity under in-vitro/in-vivo study	[72]
Cu-LDH	Hydrothermal, Isomorphic substitution	Drug delivery, phototherapy	Intercalation of indocyanine green (ICG), 5-Fluorouracil (5-FU), and surface absorption of albumin-bound paclitaxel (nAb-PTX)	-	5-FU/Cu-LDH@nAb-PTX as a pH-sensitive heat-facilitated therapeutic on-demand release nanomedicine showing strong synergy of photothermal therapy and chemotherapy in inducing apoptosis of breast cancer cells (4 T1) under 808 nm irradiated state at very low doses of 5-FU and nAb-PTX (0.25 and 0.50 mg/kg)	[73]
MnMgFe-LDH	Coprecipitation, Isomorphic substitution	Drug delivery, phototherapy	Physical adsorption of dihydroartemisinin (DHA)	Bovine serum albumin (BSA) for stabilizing agent	MnMgFe-LDH/DHA PTT), remarkable chemodynamic/photothermal therapy (CDT/PTT) synergistic effect on tumor treatment with photothermal conversion efficiency up to 10.7% without damage to normal tissues	[74]

**Table 2 nanomaterials-13-01102-t002:** Summary of LDH nanomaterials related to imaging application without any drug reported from 2019 to 2023.

	Synthetic Approaches	Types of Biomedical Applications	Active Molecules (Therapeutic Agents or/and Contrasting Agents)	Additional Agent for Stability	Remarks	Ref.
Cu-LDH	Coprecipitation, Isomorphic substitution	Fluorescent, MRI imaging	Intercalation of fluorescein isothiocyanate (FITC)	Bovine serum albumin (BSA) using a stabilizing agent for in-vivo analysis	Charge conversions in the polymer cause a pH-responsive MRI contrast capacity (T_1_-weighed magnetic resonance images) in a suitable microenvironment. According to in-vivo testing, ~4.8% of the injected dose accumulates 24-h post-injection	[78]
MgMnAl-LDH	Coprecipitation, Isomorphic substitution	MRI imaging	Substitution of Mn^2+^ on lattice and iron oxide (IO) attached to the surface	Bovine serum albumin (BSA) using a stabilizing agent for in-vivo analysis	MgMnAl-LDH@IO NP, with biocompatibility, enables accurate in-vitro and in-vivo tumor diagnosis utilizing a dual mode of MRI imaging (T_1_- and T_2_-weighted magnetic resonance signals) and contrasting agents with tumor tissues	[79]
Mn-LDH	Coprecipitation, Isomorphic substitution	MRI imaging	Substitution of Mn^2+^ on lattice and conjugation of perfluoropolyether	Bovine serum albumin (BSA) using a stabilizing agent for in-vivo analysis	Enables a potential turn on/off system with ^19^F MRI agents for the precise and specific detection of cancer diseases. The system is pH dependent, and ^19^F magnetic resonance signals are detected only in the breast tumor tissue after injecting Mn-LDH@PFPE nanoparticles	[90]
^64^CuMgAl-LDH	Coprecipitation, Hydrothermal, Isomorphic substitution	PET imaging	Substitution of ^64^Cu on lattice	Bovine serum albumin (BSA) using a stabilizing agent for in-vivo analysis	^64^Cu is immobilized in the octahedral sites of the quintinite lattice. ^64^Cu-QT-NPs/BSA, with passive targeting behavior based on EPR, visualized using a PET scanner in-vivo, shows high potential as an advanced nano-device for radio-imaging and diagnosis	[28]

**Table 3 nanomaterials-13-01102-t003:** Summary of LDH nanomaterials related to theranostics, combined drug delivery, and imaging application reported from 2019 to 2023.

	Synthetic Approaches	Types of Biomedical Applications	Active Molecules (Therapeutic Agents or/and Contrasting Agents)	Additional Agent for Stability	Remarks	Ref.
MgAl-LDH	Coprecipitation, Hydrothermal	Drug delivery & MRI imaging	Attachment of cerium oxide and iron oxide to the surface of LDHs	-	It enables the co-delivery of cerium oxide and iron oxide in combination with targeted molecules and therapeutic drugs, enabling ROS scavenging and diagnosing ROS-related diseases. It shows an excellent magnetic resonance signal without any toxicity	[127]
MnAl-LDH	Coprecipitation, Ion-exchange	Drug delivery & MRI imaging	Contains Mn^2+^ on the lattice, with intercalated fluorouracil (FU)	-	The theranostics nanoplatform FU-MnAl-LDH contains Mn as Mn^2+^ and Mn^3+^ and shows a pH-dependent FU release and MRI imaging capacity with high r_1_ and r_2_ relaxivities, particularly at pH = 7.4	[115]
MgAl-LDH	Coprecipitation, Hydrothermal	Drug delivery & Fluorescence imaging	Intercalation of flurbiprofen (FB) and fluoresce isothiocyanate (FITC) substitution	Hyaluronic acid (HA) modification for stabilizing, targeting agents	The ocular delivery system of HA-FB-LDH (74.82 ± 0.29%) has a sustained release pattern, analyzing cumulative release amounts compared to FB-LDH (92.99 ± 0.37%) within 12 h in-vitro, and improvement of the precorneal residence time in-vivo (1.48 times higher than that of the FB-LDH group)	[125]
MgAlGd-LDH	Coprecipitation	Drug delivery & MRI imaging	Intercalation of atorvastatin and partial substitution of Gd^3+^ in M^3+^ sites	Ferritin heavy subunit (FTH) as a blood-brain barrier transport agent	AFGd-LDH enables MRI imaging and ischemia-reperfusion therapy, according to its ROS scavenging efficiency in-vitro and the decrease of apoptosis induced by reperfusion in-vivo	[81]
MgAl-LDH	Coprecipitation, Hydrothermal	Drug delivery & MRI & Hyperthermia	Intercalation of methotrexate (MTX) or 5-fluorouracil (5FU) and the incorporation of iron oxide	Poly(acrylamide-co-acrylonitrile) for thermo-responsive property	Spray-dried microparticles are used for smart stimuli-responsive theranostics for hyperthermia-aided chemotherapy and MRI diagnosis. Their activity has been analyzed with tests under different temperatures, as well as in-vitro release and other cell experiments	[114]
CoAl-LDH	Exfoliation	Drug delivery & Fluorescence imaging	Intercalation of doxorubicin (DOX) and paclitaxel (PTX), RhB modification	Folic acid (FA) for better targeting of cancerous cells	It is a biodegradable and versatile drug-delivery nanocarrier based on the self-assembly of delaminated CoAl-LDHs and MnO_2_ that shows a synergistically enhanced therapeutic effect due to the co-loading of DOX and PTX. It shows better in-vitro and in-vivo activity than cocktail solutions of the two drugs	[116]
MgAl-LDH	Coprecipitation	Drug delivery & MRI	LDH is stabilized on Fe_3_O_4_, followed by the physical mixing of LDH-Fe_3_O_4_-HA with doxorubicin (DOX)	Hyaluronic acid (HA) modification for stabilizing, targeting agents	It is a novel targeted theranostics nanoplatform. LDH-Fe_3_O_4_-HA/DOX nanohybrids show efficient T_1_-weighted MRI-guided chemotherapy of CD44 receptor-overexpressing tumors	[117]
MgAl-LDH	Coprecipitation, Hydrothermal	Drug delivery & SPECT imaging	Intercalation of methotrexate (MTX) and substitution a radioisotope, Co-57, into the lattice of LDH	-	It is highly stable in human serum, and the labeled Co-57 in MTX-LDH shows a high cellular uptake with CT-26 cells in-vitro, as well as high cancer-cell suppression. It enables the visualization of in-vivo SPECT signals in tumor tissues within 1 h; the signal increases after 3 h	[29]

## Data Availability

All the data mentioned can be found in the manuscript itself. If there is any additional requirement, please contact the corresponding author.

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
