# Peer review of "Multifunctional Layered Double Hydroxides for Drug Delivery and Imaging"

_nanomaterials, 2023, doi:10.3390/nano13061102_

Round 1
Reviewer 1 Report
This review summarized one kind of two-dimensional nanomaterials used in drug delivery and bioimaging, particularly layered double hydroxides (LDHs). The whole manuscript is well-written, but it still needs to supply some key information before publication. Specific comments are listed below.
1. The author uses Lego in the scheme, please double-check if there is have copyright on this product.
2. Various weak interactions are used to synthesize bio-applicable molecule-LDH nanohybrids, please give a discussion.
3. Overcoming blood–brain barrier transport is very important in drug delivery and should be discussed for using the LDHs. And important references should be included, like Materials today 37 (2020): 112-125.
4. There are lots of reviews about 2D materials for biomedical applications, what are the specific options in this review? Also, advanced works should be highlighted, like Advanced Drug Delivery Reviews (2022): 114269.
Author Response
Please find the file as attached.

Reviewer 2 Report
The manuscript of “Multifunctional Layered Double Hydroxides for Drug Delivery and Imaging: A Mini Review” summarizes the synthetic strategies and in-vivo and/or in-vitro 23 therapeutic actions and targeting properties of single-function LDH-based nanohybrids and re-24 cently reported (from 2019 to 2023) multifunctional systems developed for drug delivery and/or bio-25 imaging. I do think this mini review is well organized and of good quality, and I believe this review will attract wide audiences, so I recommend accept this manuscript after some revisions.
(1) the mechanisms of drug delivery and imaging in LDHs were not clearly demonstrated, the authors can provide mechanisms schematic to emphasize the processes.
(2) the prospect of the LDHs nanostructure and their drug delivery and imaging should be added after the conclusion section, I do think this is necessary for a review to provide readers clear developing directions of materials and application requirements.
Author Response
Please find the file as attached.

Round 2
Reviewer 2 Report
Accept in present form